# TinyBERT: Distilling BERT for Natural Language Understanding

## Abstract

Language model pre-training, such as BERT, has significantly improved the performances of many natural language processing tasks. However, the pre-trained language models are usually computationally expensive and memory intensive, so it is difficult to effectively execute them on resource-restricted devices. To accelerate inference and reduce model size while maintaining accuracy, we firstly propose a novel *Transformer distillation* method that is specially designed for knowledge distillation (KD) of the Transformer-based models. By leveraging this new KD method, the plenty of knowledge encoded in a large "teacher" BERT can be well transferred to a small "student" TinyBERT. Moreover, we introduce a new two-stage learning framework for TinyBERT, which performs Transformer distillation at both the pre-training and task-specific learning stages. This framework ensures that TinyBERT can capture the general-domain as well as the task-specific knowledge in BERT.

TinyBERT[1] is empirically effective and achieves more than 96% the performance of teacher BERT$_{\text{BASE}}$ on GLUE benchmark, while being **7.5x smaller** and **9.4x faster** on inference. TinyBERT is also significantly better than state-of-the-art baselines on BERT distillation, with only $\sim$**28%** parameters and $\sim$**31%** inference time of them.

## 1 Introduction

Pre-training language models then fine-tuning on downstream tasks has become a new paradigm for natural language processing (NLP). Pre-trained language models (PLMs), such as BERT (Devlin et al., 2018), XLNet (Yang et al., 2019), RoBERTa (Liu et al., 2019) and SpanBERT (Joshi et al., 2019), have achieved great success in many NLP tasks (e.g., the GLUE benchmark (Wang et al., 2018) and the challenging multi-hop reasoning task (Ding et al., 2019)). However, PLMs usually have an extremely large number of parameters and need long inference time, which are difficult to be deployed on edge devices such as mobile phones. Moreover, recent studies (Kovaleva et al., 2019) also demonstrate that there is redundancy in PLMs. Therefore, it is crucial and possible to reduce the computational overhead and model storage of PLMs while keeping their performances.

There has been many model compression techniques (Han et al., 2015a) proposed to accelerate deep model inference and reduce model size while maintaining accuracy. The most commonly used techniques include quantization (Gong et al., 2014), weights pruning (Han et al., 2015b), and knowledge distillation (KD) (Romero et al., 2014). In this paper we focus on knowledge distillation, an idea proposed by Hinton et al. (2015) in a *teacher-student* framework. KD aims to transfer the knowledge embedded in a large teacher network to a small student network. The student network is trained to reproduce the behaviors of the teacher network. Based on the framework, we propose a novel distillation method specifically for Transformer-based models (Vaswani et al., 2017), and use BERT as an example to investigate the KD methods for large scale PLMs.

KD has been extensively studied in NLP (Kim & Rush, 2016; Hu et al., 2018), while designing KD methods for BERT has been less explored. The *pre-training-then-fine-tuning* paradigm firstly pre-trains BERT on a large scale unsupervised text corpus, then fine-tunes it on task-specific dataset, which greatly increases the difficulty of BERT distillation. Thus we are required to design an ef-

---

[1]Our code and models will be made publicly available.

fective KD strategy for both stages. To build a competitive TinyBERT, we firstly propose a new *Transformer distillation* method to distill the knowledge embedded in teacher BERT. Specifically, we design several loss functions to fit different representations from BERT layers: 1) the output of the embedding layer; 2) the hidden states and attention matrices derived from the Transformer layer; 3) the logits output by the prediction layer. The attention based fitting is inspired by the recent findings (Clark et al., 2019) that the attention weights learned by BERT can capture substantial linguistic knowledge, which encourages that the linguistic knowledge can be well transferred from teacher BERT to student TinyBERT. However, it is ignored in existing KD methods of BERT, such as Distilled BiLSTM$_{SOFT}$ (Tang et al., 2019), BERT-PKD (Sun et al., 2019) and DistilBERT[2]. Then, we propose a novel *two-stage learning* framework including the *general distillation* and the *task-specific distillation*. At the general distillation stage, the original BERT without fine-tuning acts as the teacher model. The student TinyBERT learns to mimic the teacher's behavior by executing the proposed Transformer distillation on the large scale corpus from general domain. We obtain a general TinyBERT that can be fine-tuned for various downstream tasks. At the task-specific distillation stage, we perform the data augmentation to provide more task-specific data for teacher-student learning, and then re-execute the Transformer distillation on the augmented data. Both the two stages are essential to improve the performance and generalization capability of TinyBERT. A detailed comparison between the proposed method and other existing methods is summarized in Table 1. The Transformer distillation and two-stage learning framework are two key ideas of the proposed method.

Table 1: A summary of KD methods for BERT. Abbreviations: INIT(initializing student BERT with some layers of pre-trained teacher BERT), DA(conducting data augmentation for task-specific training data). Embd, Attn, Hidn, and Pred represent the knowledge from embedding layers, attention matrices, hidden states, and final prediction layers, respectively.

| KD Methods | KD at Pre-training Stage | | | | | KD at Fine-tuning Stage | | | | |
|---|---|---|---|---|---|---|---|---|---|---|
| | INIT | Embd | Attn | Hidn | Pred | Embd | Attn | Hidn | Pred | DA |
| Distilled BiLSTM$_{SOFT}$ | | | | | | | | | ✓ | ✓ |
| BERT-PKD | ✓ | | | | | | | ✓[3] | ✓ | |
| DistilBERT | ✓ | | | | ✓[4] | | | | ✓ | |
| TinyBERT (our method) | | ✓ | ✓ | ✓ | | ✓ | ✓ | ✓ | ✓ | ✓ |

The main contributions of this work are as follows: 1) We propose a new Transformer distillation method to encourage that the linguistic knowledge encoded in teacher BERT can be well transferred to TinyBERT. 2) We propose a novel two-stage learning framework with performing the proposed Transformer distillation at both the pre-training and fine-tuning stages, which ensures that Tiny-BERT can capture both the general-domain and task-specific knowledge of the teacher BERT. 3) We show experimentally that our TinyBERT can achieve more than 96% the performance of teacher BERT$_{BASE}$ on GLUE tasks, while having much fewer parameters (∼13.3%) and less inference time (∼10.6%), and significantly outperforms other state-of-the-art baselines on BERT distillation.

## 2 PRELIMINARIES

We firstly describe the formulation of Transformer (Vaswani et al., 2017) and Knowledge Distillation (Hinton et al., 2015). Our proposed Transformer distillation is a specially designed KD method for Transformer-based models.

### 2.1 TRANSFORMER LAYER

Most of the recent pre-trained language models (e.g., BERT, XLNet and RoBERTa) are built with Transformer layers, which can capture long-term dependencies between input tokens by self-attention mechanism. Specifically, a standard Transformer layer includes two main sub-layers: *multi-head attention* (MHA) and *fully connected feed-forward* network (FFN).

---

[2]https://medium.com/huggingface/distilbert-8cf3380435b5

[3]The student learns from the `[CLS]` (a special classification token of BERT) hidden states of the teacher.

[4]The output of pre-training tasks (such as dynamic masking) is used as the supervision signal.

**Multi-Head Attention (MHA)**. The calculation of attention function depends on the three components of queries, keys and values, which are denoted as matrices $\boldsymbol{Q}$, $\boldsymbol{K}$ and $\boldsymbol{V}$ respectively. The attention function can be formulated as follows:

$$A = \frac{\boldsymbol{Q}\boldsymbol{K}^T}{\sqrt{d_k}}, \tag{1}$$

$$\texttt{Attention}(\boldsymbol{Q}, \boldsymbol{K}, \boldsymbol{V}) = \texttt{softmax}(\boldsymbol{A})\boldsymbol{V}, \tag{2}$$

where $d_k$ is the dimension of keys and acts as a scaling factor, $\boldsymbol{A}$ is the attention matrix calculated from the compatibility of $\boldsymbol{Q}$ and $\boldsymbol{K}$ by dot-product operation. The final function output is calculated as a weighted sum of values $\boldsymbol{V}$, and the weight is computed by applying $\texttt{softmax()}$ operation on the each column of matrix $\boldsymbol{A}$. According to Clark et al. (2019), the attention matrices in BERT can capture substantial linguistic knowledge, and thus play an essential role in our proposed distillation method.

Multi-head attention is defined by concatenating the attention heads from different representation subspaces as follows:

$$\texttt{MultiHead}(\boldsymbol{Q}, \boldsymbol{K}, \boldsymbol{V}) = \texttt{Concat}(\text{head}_1, \ldots, \text{head}_h)\boldsymbol{W}, \tag{3}$$

where $h$ is the number of attention heads, and $\text{head}_i$ denotes the $i$-th attention head, which is calculated by the $\texttt{Attention()}$ function with inputs from different representation subspaces, the matrix $\boldsymbol{W}$ acts as a linear transformation.

**Position-wise Feed-Forward Network (FFN)**. Transformer layer also contains a fully connected feed-forward network, which is formulated as follows:

$$\texttt{FNN}(x) = \max(0, x\boldsymbol{W}_1 + b_1)\boldsymbol{W}_2 + b_2. \tag{4}$$

We can see that the FFN contains two linear transformations and one ReLU activation.

## 2.2 Knowledge Distillation

KD aims to transfer the knowledge of a large teacher network $T$ to a small student network $S$. The student network is trained to mimic the behaviors of teacher networks. Let $f^T$ and $f^S$ represent the *behavior* functions of teacher and student networks, respectively. The behavior function targets at transforming network inputs to some informative representations, and it can be defined as the output of any layer in the network. In the context of Transformer distillation, the output of MHA layer or FFN layer, or some intermediate representations (such as the attention matrix $\boldsymbol{A}$) can be used as behavior function. Formally, KD can be modeled as minimizing the following objective function:

$$\mathcal{L}_{\text{KD}} = \sum_{x \in \mathcal{X}} L\big(f^S(x), f^T(x)\big), \tag{5}$$

where $L(\cdot)$ is a loss function that evaluates the difference between teacher and student networks, $x$ is the text input and $\mathcal{X}$ denotes the training dataset. Thus the key research problem becomes how to define effective behavior functions and loss functions. Different from previous KD methods, we also need to consider how to perform KD at the pre-training stage of BERT in addition to the task-specific training stage.

## 3 Method

In this section, we propose a novel distillation method for Transformer-based models, and present a two-stage learning framework for our model distilled from BERT, which is called TinyBERT.

### 3.1 Transformer Distillation

The proposed *Transformer distillation* is a specially designed KD method for Transformer networks. Figure 1 displays an overview of the proposed KD method. In this work, both the student and teacher networks are built with Transformer layers. For a clear illustration, we firstly formulate the problem before introducing our method.

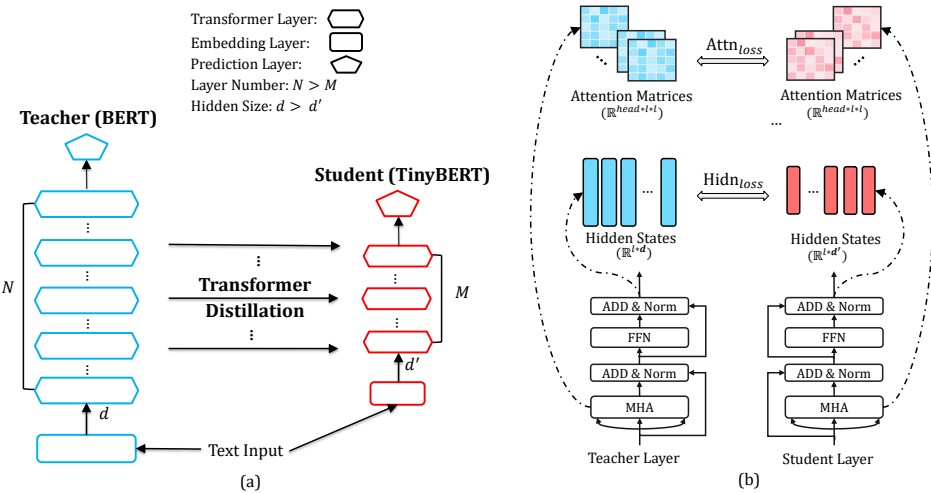

Figure 1: An overview of Transformer distillation: (a) the framework of Transformer distillation, (b) the details of Transformer-layer distillation consisting of $\text{Attn}_{loss}$(attention based distillation) and $\text{Hidn}_{loss}$(hidden states based distillation).

**Problem Formulation**. Assuming that the student model has $M$ Transformer layers and teacher model has $N$ Transformer layers, we choose $M$ layers from the teacher model for the *Transformer-layer distillation*. The function $n = g(m)$ is used as a mapping function from student layers to teacher layers, which means that the $m$-th layer of student model learns the information from the $n$-th layer of teacher model. The *embedding-layer distillation* and the *prediction-layer distillation* are also considered. We set 0 to be the index of embedding layer and $M + 1$ to be the index of prediction layer, and the corresponding layer mappings are defined as $0 = g(0)$ and $N + 1 = g(M + 1)$ respectively. The effect of the choice of different mapping functions on the performances will be studied in the experiment section. Formally, the student can acquire knowledge from the teacher by minimizing the following objective:

$$\mathcal{L}_{\text{model}} = \sum_{m=0}^{M+1} \lambda_m \mathcal{L}_{\text{layer}}(S_m, T_{g(m)}), \qquad (6)$$

where $\mathcal{L}_{\text{layer}}$ refers to the loss function of a given model layer (e.g., Transformer layer or embedding layer) and $\lambda_m$ is the hyper-parameter that represents the importance of the $m$-th layer's distillation.

**Transformer-layer Distillation**. The proposed Transformer-layer distillation includes the *attention based distillation* and *hidden states based distillation*, which is shown in Figure 1 (b). The attention based distillation is motivated by the recent findings that attention weights learned by BERT can capture rich linguistic knowledge (Clark et al., 2019). This kind of linguistic knowledge includes the syntax and coreference information, which is essential for natural language understanding. Thus we propose the attention based distillation to encourage that the linguistic knowledge can be transferred from teacher BERT to student TinyBERT. Specifically, the student learns to fit the matrices of multi-head attention in the teacher network, and the objective is defined as:

$$\mathcal{L}_{\text{attn}} = \frac{1}{h} \sum_{i=1}^{h} \text{MSE}(\boldsymbol{A}_i^S, \boldsymbol{A}_i^T), \qquad (7)$$

where $h$ is the number of attention heads, $\boldsymbol{A}_i \in \mathbb{R}^{l \times l}$ refers to the attention matrix corresponding to the $i$-th head of teacher or student, $l$ is the input text length, and $\text{MSE}()$ means the *mean squared error* loss function. In this work, the (unnormalized) attention matrix $\boldsymbol{A}_i$ is used as the fitting target instead of its softmax output $\text{softmax}(\boldsymbol{A}_i)$, since our experiments show that the former setting has a faster convergence rate and better performances.

In addition to the attention based distillation, we also distill the knowledge from the output of Transformer layer (as shown in Figure 1 (b)), and the objective is as follows:

$$\mathcal{L}_{\text{hidn}} = \text{MSE}(\boldsymbol{H}^S \boldsymbol{W}_h, \boldsymbol{H}^T), \qquad (8)$$

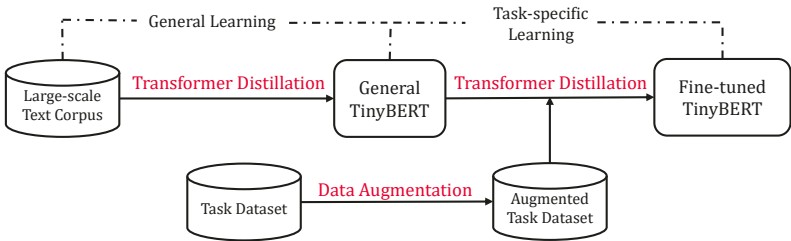

Figure 2: The illustration of TinyBERT learning

where the matrices $\boldsymbol{H}^S \in \mathbb{R}^{l \times d'}$ and $\boldsymbol{H}^T \in \mathbb{R}^{l \times d}$ refer to the hidden states of student and teacher networks respectively, which are calculated by Equation 4. The scalar values $d$ and $d'$ denote the hidden sizes of teacher and student models, and $d'$ is often smaller than $d$ to obtain a smaller student network. The matrix $\boldsymbol{W}_h \in \mathbb{R}^{d' \times d}$ is a learnable linear transformation, which transforms the hidden states of student network into the same space as the teacher network's states.

**Embedding-layer Distillation**. We also perform embedding-layer distillation, which is similar to the hidden states based distillation and formulated as:

$$\mathcal{L}_{\text{embd}} = \texttt{MSE}(\boldsymbol{E}^S \boldsymbol{W}_e, \boldsymbol{E}^T), \tag{9}$$

where the matrices $\boldsymbol{E}^S$ and $\boldsymbol{H}^T$ refer to the embeddings of student and teacher networks, respectively. In this paper, they have the same shape as the hidden state matrices. The matrix $\boldsymbol{W}_e$ is a linear transformation playing a similar role as $\boldsymbol{W}_h$.

**Prediction-Layer Distillation**. In addition to imitating the behaviors of intermediate layers, we also use the knowledge distillation to fit the predictions of teacher model (Hinton et al., 2015). Specifically, we penalize the soft cross-entropy loss between the student network's logits against the teacher's logits:

$$\mathcal{L}_{\text{pred}} = -\texttt{softmax}(\boldsymbol{z}^T) \cdot \texttt{log\_softmax}(\boldsymbol{z}^S / t), \tag{10}$$

where $\boldsymbol{z}^S$ and $\boldsymbol{z}^T$ are the logits vectors predicted by the student and teacher respectively, $\texttt{log\_softmax}()$ means the log likelihood, $t$ means the temperature value. In our experiment, we find that $t = 1$ performs well.

Using the above distillation objectives (i.e. Equations 7, 8, 9 and 10), we can unify the distillation loss of the corresponding layers between the teacher and the student network:

$$\mathcal{L}_{\text{layer}}(S_m, T_{g(m)}) = \begin{cases} \mathcal{L}_{\text{embd}}(S_0, T_0), & m = 0 \\ \mathcal{L}_{\text{hidn}}(S_m, T_{g(m)}) + \mathcal{L}_{\text{attn}}(S_m, T_{g(m)}), & M \geq m > 0 \\ \mathcal{L}_{\text{pred}}(S_{M+1}, T_{N+1}), & m = M + 1 \end{cases} \tag{11}$$

In our experiments, we firstly perform *intermediate layer distillation* ($M \geq m \geq 0$), then perform the *prediction-layer distillation* ($m = M + 1$).

## 3.2 TinyBERT Learning

The application of BERT usually consists of two learning stages: the pre-training and fine-tuning. The plenty of knowledge learned by BERT in the pre-training stage is of great importance and should also be transferred. Therefore, we propose a novel two-stage learning framework including the *general distillation* and the *task-specific distillation*, as illustrated in Figure 2. General distillation helps student TinyBERT learn the rich knowledge embedded in teacher BERT, which plays an important role in improving the generalization capability of TinyBERT. The task-specific distillation teaches the student the task-specific knowledge. With the two-step distillation, we can further reduce the gap between teacher and student models.

**General Distillation**. In general distillation, we use the original BERT without fine-tuning as the teacher and a large-scale text corpus as the learning data. By performing the Transformer distillation[5] on the text from general domain, we obtain a general TinyBERT that can be fine-tuned for

---

[5]In the general distillation, we do not perform prediction-layer distillation as Equation 10.

Table 2: Results are evaluated on the test set of GLUE official benchmark. All models are learned in a single-task manner. "-" means the result is not reported.

| System | MNLI-m | MNLI-mm | QQP | SST-2 | QNLI | MRPC | RTE | CoLA | STS-B | Average |
|---|---|---|---|---|---|---|---|---|---|---|
| BERT$_{\text{BASE}}$ (Google) | 84.6 | 83.4 | 71.2 | 93.5 | 90.5 | 88.9 | 66.4 | 52.1 | 85.8 | 79.6 |
| BERT$_{\text{BASE}}$ (Teacher) | 83.9 | 83.4 | 71.1 | 93.4 | 90.9 | 87.5 | 67.0 | 52.8 | 85.2 | 79.5 |
| BERT$_{\text{SMALL}}$ | 75.4 | 74.9 | 66.5 | 87.6 | 84.8 | 83.2 | 62.6 | 19.5 | 77.1 | 70.2 |
| Distilled BiLSTM$_{\text{SOFT}}$ | 73.0 | 72.6 | 68.2 | 90.7 | - | - | - | - | - | - |
| BERT-PKD | 79.9 | 79.3 | 70.2 | 89.4 | 85.1 | 82.6 | 62.3 | 24.8 | 79.8 | 72.6 |
| DistilBERT | 78.9 | 78.0 | 68.5 | 91.4 | 85.2 | 82.4 | 54.1 | 32.8 | 76.1 | 71.9 |
| TinyBERT | 82.5 | 81.8 | 71.3 | 92.6 | 87.7 | 86.4 | 62.9 | 43.3 | 79.9 | 76.5 |

Table 3: The model sizes and inference time for baselines and TinyBERT. The number of layers does not include the embedding and prediction layers.

| System | Layers | Hidden Size | Feed-forward Size | Model Size | Inference Time |
|---|---|---|---|---|---|
| BERT$_{\text{BASE}}$ (Teacher) | 12 | 768 | 3072 | 109M($\times$1.0) | 188s($\times$1.0) |
| Distilled BiLSTM$_{\text{SOFT}}$ | 1 | 300 | 400 | 10.1M($\times$10.8) | 24.8s($\times$7.6) |
| BERT-PKD/DistilBERT | 4 | 768 | 3072 | 52.2M($\times$2.1) | 63.7s($\times$3.0) |
| TinyBERT/BERT$_{\text{SMALL}}$ | 4 | 312 | 1200 | 14.5M($\times$7.5) | 19.9s($\times$9.4) |

downstream tasks. However, due to the significant reductions of the hidden/embedding size and the layer number, general TinyBERT performs relatively worse than BERT.

**Task-specific Distillation**. Previous studies show that the complex models, fine-tuned BERTs, suffer from over-parametrization for domain-specific tasks (Kovaleva et al., 2019). Thus, it is possible for small models to achieve comparable performances to the BERTs. To this end, we propose to derive competitive fine-tuned TinyBERTs through the task-specific distillation. In the task-specific distillation, we re-perform the proposed Transformer distillation on an augmented task-specific dataset (as shown in Figure 2). Specifically, the fine-tuned BERT is used as the teacher and a data augmentation method is proposed to expand the task-specific training set. Learning more task-related examples, the generalization capabilities of student model can be further improved. In this work, we combine a pre-trained language model BERT and GloVe (Pennington et al., 2014) word embeddings to do word-level replacement for data augmentation. Specifically, we use the language model to predict word replacements for single-piece words (Wu et al., 2019), and use the word embeddings to retrieve the most similar words as word replacements for multiple-pieces words. Some hyper-parameters are defined to control the replacement ratio of a sentence and the amount of augmented dataset. More details of the data augmentation procedure are discussed in Appendix A.

The above two learning stages are complementary to each other: the general distillation provides a good initialization for the task-specific distillation, while the task-specific distillation further improves TinyBERT by focusing on learning the task-specific knowledge. Although there is a big gap between BERT and TinyBERT in model size, by performing the proposed two-stage distillation, the TinyBERT can achieve competitive performances in various NLP tasks. The proposed *Transformer distillation* and *two-stage learning framework* are the two most important components of the proposed distillation method.

# 4 EXPERIMENTS

In this section, we evaluate the effectiveness and efficiency of TinyBERT on a variety of tasks with different model settings.

## 4.1 MODEL SETUP

We instantiate a tiny student model (the number of layers $M$=4, the hidden size $d'$=312, the feed-forward/filter size $d'_i$=1200 and the head number $h$=12) that has a total of 14.5M parameters. If not specified, this student model is referred to as the TinyBERT. The original BERT$_{\text{BASE}}$ (the number of layers $N$=12, the hidden size $d$=768, the feed-forward/filter size $d_i$=3072 and the head number $h$=12) is used as the teacher model that contains 109M parameters. We use $g(m) = 3 \times m$ as the layer mapping function, so TinyBERT learns from every 3 layers of BERT$_{\text{BASE}}$. The learning weight $\lambda$ of each layer is set to 1, which performs well for the learning of our TinyBERT.

Table 4: Results (dev) of wider or deeper TinyBERT variants and baselines.

| System | MNLI-m | MNLI-mm | MRPC | CoLA | Average |
|---|---|---|---|---|---|
| $\text{BERT}_{\text{BASE}}$ (Teacher) | 84.2 | 84.4 | 86.8 | 57.4 | 78.2 |
| BERT-PKD ($M$=6;$d'$=768;$d'_i$=3072) | 80.9 | 80.9 | 83.1 | 43.1 | 72.0 |
| DistilBERT ($M$=6;$d'$=768;$d'_i$=3072) | 81.6 | 81.1 | 82.4 | 42.5 | 71.9 |
| TinyBERT ($M$=4;$d'$=312;$d'_i$=1200) | 82.8 | 82.9 | 85.8 | 49.7 | 75.3 |
| TinyBERT ($M$=4;$d'$=768;$d'_i$=3072) | 83.8 | 84.1 | 85.8 | 50.5 | 76.1 |
| TinyBERT ($M$=6;$d'$=312;$d'_i$=1200) | 83.3 | 84.0 | 86.3 | 50.6 | 76.1 |
| TinyBERT ($M$=6;$d'$=768;$d'_i$=3072) | 84.5 | 84.5 | 86.3 | 54.0 | 77.3 |

## 4.2 EXPERIMENTAL RESULTS ON GLUE

We evaluate TinyBERT on the General Language Understanding Evaluation (GLUE) (Wang et al., 2018) benchmark, which is a collection of diverse natural language understanding tasks. The details of experiment settings are described in Appendix B. The evaluation results are presented in Table 2 and the efficiencies of model size and inference time are also evaluated in Table 3.

The experiment results demonstrate that: 1) There is a large performance gap between $\text{BERT}_{\text{SMALL}}$[6] and $\text{BERT}_{\text{BASE}}$ due to the big reduction in model size. 2) TinyBERT is consistently better than $\text{BERT}_{\text{SMALL}}$ in all the GLUE tasks and achieves a large improvement of 6.3% on average. This indicates that the proposed KD learning framework can effectively improve the performances of small models regardless of downstream tasks. 3) TinyBERT significantly outperforms the state-of-the-art KD baselines (i.e., BERT-PKD and DistillBERT) by a margin of at least 3.9%, even with only ∼28% parameters and ∼31% inference time of baselines (see Table 3). 4) Compared with the teacher $\text{BERT}_{\text{BASE}}$, TinyBERT is 7.5x smaller and 9.4x faster in the model efficiency, while maintaining competitive performances. 5) TinyBERT has a comparable model efficiency (slightly larger in size but faster in inference) with Distilled $\text{BiLSTM}_{\text{SOFT}}$ and obtains substantially better performances in all tasks reported by the BiLSTM baseline. 6) For the challenging CoLA dataset (the task of predicting linguistic acceptability judgments), all the distilled small models have a relatively bigger performance gap with teacher model. TinyBERT achieves a significant improvement over the strong baselines, and its performance can be further improved by using a deeper and wider model to capture more complex linguistic knowledge as illustrated in the next subsection.

Moreover, BERT-PKD and DistillBERT initialize their student models with some layers of well pre-trained teacher BERT (see Table 1), which makes the student models have to keep the same size settings of Transformer layer (or embedding layer) as their teacher BERT. In our two-stage distillation framework, TinyBERT is initialized by general distillation, so it has the advantage of being more flexible in model size selection.

## 4.3 EFFECTS OF MODEL SIZE

We evaluate how much improvement can be achieved when increasing the model size of TinyBERT on several typical GLUE tasks, where MNLI and MRPC are used in the ablation studies of Devlin et al. (2018), and CoLA is the most difficult task in GLUE. Specifically, three wider and deeper variants are proposed and their evaluation results on development set are displayed in Table 4. We can observe that: 1) All the three TinyBERT variants can consistently outperform the original s-mallest TinyBERT, which indicates that the proposed KD method works for the student models of various model sizes. 2) For the CoLA task, the improvement is slight when only increasing the number of layers (from 49.7 to 50.6) or hidden size (from 49.7 to 50.5). To achieve more dramatic improvements, the student model should become deeper and wider (from 49.7 to 54.0). 3) Another interesting observation is that the smallest 4-layer TinyBERT can even outperform the 6-layers baselines, which further confirms the effectiveness of the proposed KD method.

## 4.4 ABLATION STUDIES

In this section, we conduct ablation studies to investigate the contributions of : 1) different procedures of the proposed two-stage TinyBERT learning framework (see Figure 2), and 2) different distillation objectives (see Equation 11).

---

[6]$\text{BERT}_{\text{SMALL}}$ means directly pretraining a small BERT, which has the same model architecture as TinyBERT, through tasks of *Masked Language Model* (MLM) and *Next Sentence Prediction* (NSP).

Table 5: Ablation studies of different procedures (i.e., TD, GD, and DA) of the two-stage learning framework. The variants are validated on the dev set.

| System | MNLI-m | MNLI-mm | MRPC | CoLA | Average |
|---|---|---|---|---|---|
| TinyBERT | 82.8 | 82.9 | 85.8 | 49.7 | 75.3 |
| No GD | 82.5 | 82.6 | 84.1 | 40.8 | 72.5 |
| No TD | 80.6 | 81.2 | 83.8 | 28.5 | 68.5 |
| No DA | 80.5 | 81.0 | 82.4 | 29.8 | 68.4 |

Table 6: Ablation studies of different distillation objectives in the TinyBERT learning. The variants are validated on the dev set.

| System | MNLI-m | MNLI-mm | MRPC | CoLA | Average |
|---|---|---|---|---|---|
| TinyBERT | 82.8 | 82.9 | 85.8 | 49.7 | 75.3 |
| No Embd | 82.3 | 82.3 | 85.0 | 46.7 | 74.1 |
| No Pred | 80.5 | 81.0 | 84.3 | 48.2 | 73.5 |
| No Trm | 71.7 | 72.3 | 70.1 | 11.2 | 56.3 |
| No Attn | 79.9 | 80.7 | 82.3 | 41.1 | 71.0 |
| No Hidn | 81.7 | 82.1 | 84.1 | 43.7 | 72.9 |

Table 7: Results (dev) of different mapping strategies.

| System | MNLI-m | MNLI-mm | MRPC | CoLA | Average |
|---|---|---|---|---|---|
| TinyBERT (Uniform-strategy) | 82.8 | 82.9 | 85.8 | 49.7 | 75.3 |
| TinyBERT (Top-strategy) | 81.7 | 82.3 | 83.6 | 35.9 | 70.9 |
| TinyBERT (Bottom-strategy) | 80.6 | 81.3 | 84.6 | 38.5 | 71.3 |

**Effects of different learning procedures.** The proposed two-stage TinyBERT learning framework (see Figure 2) consists of three key procedures: TD (Task-specific Distillation), GD (General Distillation) and DA (Data Augmentation). The effects of different learning procedures are analyzed and presented in Table 5. The results indicate that all the three procedures are crucial for the proposed KD method. The TD and DA has comparable effects in all the four tasks. We can also find the task-specific procedures (TD and DA) are more helpful than the pre-training procedure (GD) in all the four tasks. Another interesting observation is that GD has more effect on CoLA than on MNLI and MRPC. We conjecture that the ability of linguistic generalization (Warstadt et al., 2018) learned by GD plays a more important role in the downstream CoLA task (linguistic acceptability judgments).

**Effects of different distillation objectives.** We investigate the effects of distillation objectives on the TinyBERT learning. Several baselines are proposed including the TinyBERT learning without the Transformer-layer distillation (No Trm), embedding-layer distillation (No Emb) and prediction-layer distillation (No Pred)[7] respectively. The results are illustrated in Table 6 and show that all the proposed distillation objectives are useful for the TinyBERT learning. The performance drops significantly from 75.3 to 56.3 under the setting (No Trm), which indicates Transformer-layer distillation is the key for TinyBERT learning. Furthermore, we study the contributions of attention (No Attn) and hidden states (No Hidn) in the Transformer-layer distillation. We can find the attention based distillation has a bigger effect than hidden states based distillation on TinyBERT learning. Meanwhile, these two kinds of knowledge distillation are complementary to each other, which makes TinyBERT obtain the competitive results.

## 4.5 EFFECTS OF MAPPING FUNCTION

We investigate the effects of different mapping functions $n = g(m)$ on the TinyBERT learning. Our original TinyBERT as described in section 4.1 uses the uniform-strategy, and we compare with two typical baselines including top-strategy $(g(m) = m + N - M; 0 < m \leq M)$ and bottom-strategy $(g(m) = m; 0 < m \leq M)$.

The comparison results are presented in Table 7. We find that the top-strategy performs better than the bottom-strategy in MNLI, while being worse in MRPC and CoLA tasks, which confirms the observations that different tasks depend on the different kinds of knowledge from BERT layers. Since the uniform-strategy acquires the knowledge from bottom to top layers of $BERT_{BASE}$, it achieves better performances than the other two baselines in all the four tasks. Adaptively choosing layers for a specific task is a challenging problem and we leave it as the future work.

**Other Experiments.** We also evaluate TinyBERT on the question answering tasks, and study whether we can use $BERT_{SMALL}$ as the initialization of the general TinyBERT. The experiments are detailed in Appendix C and D.

---

[7]The prediction-layer distillation performs soft cross-entropy as Equation 10 on the augmented training set. "No Pred" means performing standard cross-entropy against the ground-truth of the original training set.

## 5 CONCLUSION AND FUTURE WORK

In this paper, we firstly introduce a new KD method for Transformer-based distillation, then we further propose a two-stage framework for TinyBERT learning. Extensive experiments show that the TinyBERT achieves competitive performances meanwhile significantly reducing the model size and shortening the inference time of original BERT$_{BASE}$, which provides an effective way to deploy BERT-based NLP applications on the edge devices.

In future work, we would study how to effectively transfer the knowledge from wider and deeper teachers (e.g., BERT$_{LARGE}$ and XLNet$_{LARGE}$) to student TinyBERT. The joint learning of distillation and quantization/pruning would be another promising direction to further compress the pre-trained language models.

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

APPENDIX

# A    DATA AUGMENTATION DETAILS

In this section, we explain the proposed data augmentation method. Specifically, we firstly mask a word in a sentence, then use BERT as a language model to predict $M$ most-likely words in the corresponding position, while keeping other words unchanged. By this way, we can easily get the candidates for each word under a specific context. To induce new instances for a given sentence, we also use a threshold $p_t$ to determine whether we should replace the current word with a randomly selected candidate. By repetitively performing this replacement operation for each word in a sentence, we can finally get a new augmented sentence. In our preliminary experiments, we find the quality of generated candidates for the words consisting of multiple sub-word pieces, is relatively low. To alleviate this problem, we instead pick a similar word from GloVe (Pennington et al., 2014) word embeddings based on the cosine similarity. We apply this data augmentation method $N$ times to all the sentences of a downstream task. In this work, we set $p_t = 0.4$, $N = 20$, $M = 15$ for all our experiments. The data augmentation procedure is illustrated as below:

---

**Algorithm 1** The Proposed Data Augmentation
**Input**: $\mathbf{x}$ is a sequence of words
        $p_t, N, M$ are hyperparameters
**Output**: $data\_aug$, the augmented data

---

 1: **function** DATA_AUGMENTANTION($\mathbf{x}, p_t, N$)
 2:      $n \leftarrow 0$
 3:      $data\_aug \leftarrow [\,]$
 4:      **while** $n < N$ **do**
 5:          $\mathbf{x}_{masked} \leftarrow \mathbf{x}$
 6:          **for** $i \leftarrow 1\ to\ \texttt{len}(\mathbf{x})$ **do**
 7:              **if** $\mathbf{x}[i]$ is a single-piece word **then**
 8:                  Replace $\mathbf{x}_{masked}[i]$ with $\texttt{[MASK]}$
 9:                  $candidates \leftarrow M$ most-likely words predicted by $\texttt{BertModel}(\mathbf{x}_{masked})[i]$
10:              **else**
11:                  $candidates \leftarrow M$ similar words of $\mathbf{x}[i]$ from GloVe
12:              **end if**
13:              Sample $p \sim \texttt{UNIFORM}(0, 1)$
14:              **if** $p \leq p_t$ **then**
15:                  Replace $\mathbf{x}_{masked}[i]$ with a word from $candidates$ randomly
16:              **else**
17:                  Keep $\mathbf{x}_{masked}[i]$ as $\mathbf{x}[i]$ unchanged
18:              **end if**
19:          **end for**
20:          Append $\mathbf{x}_{masked}$ to $data\_aug$
21:          $n = n + 1$
22:      **end while**
23:      **return** $data\_aug$
24: **end function**

---

# B    GLUE DETAILS

**TinyBERT setup.** TinyBERT learning includes the general distillation and the task-specific distillation. For the general distillation, we use English Wikipedia (2,500 M words) as the text corpus and perform the *intermediate layer distillation* for 3 epochs with the supervision from a pre-trained BERT$_{BASE}$ teacher and keep other hyper-parameters same as BERT pre-training (Devlin et al., 2018). For the task-specific distillation, we firstly perform *intermediate layer distillation* on the augmented dataset for 10 epochs with batch size 32 and learning rate 5e-5 under the supervision of a fine-tuned BERT teacher, and then perform *prediction layer distillation* for 3 epochs with batch size 32 and learning rate 3e-5. For tasks like MNLI, QQP and QNLI which have $\geq 100K$ training ex-

amples, we distill intermediate layer knowledge for 5 epochs with batch size 256 on the augmented dataset. Besides, for CoLA task, we perform 50 epochs of *intermediate layer distillation*.

**Baselines setup.** We use BERT-PKD and DistilBERT as our baselines. For a fair comparison, we firstly re-implemented the results of BERT-PKD and DistilBERT reported in their papers to ensure our implementation procedure is correct. Then following the verified implementation procedure, we trained a 4-layer BERT-PKD and a 4-layer DistilBERT as the baselines. The $BERT_{SMALL}$ learning strictly follows the same learning strategy as described in the original BERT work (Devlin et al., 2018).

**Model efficiency evaluation.** To evaluate the inference speed, we ran inference procedure on the QNLI training set with batch size of 128 and the maximum sequence length of 128. The numbers reported in Table 3 are the average running time of 100 batches on a single NVIDIA K80 GPU.

The GLUE datasets are described as follows:

**MNLI.** Multi-Genre Natural Language Inference is a large-scale, crowd-sourced entailment classification task (Williams et al., 2018). Given a pair of $\langle premise, hypothesis \rangle$, the goal is to predict whether the $hypothesis$ is an entailment, contradiction, or neutral with respect to the $premise$.

**QQP.** Quora Question Pairs is a collection of question pairs from the website Quora. The task is to determine whether two questions are semantically equivalent (Chen et al., 2018).

**QNLI.** Question Natural Language Inference is a version of the Stanford Question Answering Dataset which has been converted to a binary sentence pair classification task by Wang et al. (2018). Given a pair $\langle question, context \rangle$. The task is to determine whether the $context$ contains the answer to the $question$.

**SST-2.** The Stanford Sentiment Treebank is a binary single-sentence classification task, where the goal is to predict the sentiment of movie reviews (Socher et al., 2013).

**CoLA.** The Corpus of Linguistic Acceptability is a task to predict whether an English sentence is a grammatically correct one (Warstadt et al., 2018).

**STS-B.** The Semantic Textual Similarity Benchmark is a collection of sentence pairs drawn from news headlines and many other domains (Cer et al., 2017). The task aims to evaluate how similar two pieces of texts are by a score from 1 to 5.

**MRPC.** Microsoft Research Paraphrase Corpus is a paraphrase identification dataset where systems aim to identify if two sentences are paraphrases of each other (Dolan & Brockett, 2005).

**RTE.** Recognizing Textual Entailment is a binary entailment task with a small training dataset (Bentivogli et al., 2009).

## C SQuAD 1.1 AND 2.0

We also demonstrate the effectiveness of TinyBERT on the question answering (QA) tasks: SQuAD v1.1 (Rajpurkar et al., 2016) and v2.0 (Rajpurkar et al., 2018). Following the learning procedure in the previous work (Devlin et al., 2018), we treat these two tasks as the problem of sequence labeling which predicts the possibility of each token as the start or end of answer span. We follow the settings of task-specific distillation in GLUE tasks, except with 3 running epochs and a learning rate of 5e-5 for the prediction-layer distillation on the original training dataset. The results are shown in Table 8.

The results show that TinyBERT consistently outperforms the baselines in both the small and medium size, which indicates that the proposed framework also works for the tasks of token-level labeling. Compared with sequence-level GLUE tasks, the question answering tasks depends on more subtle knowledge to infer the correct answer, which increases the difficulty of knowledge distillation. We leave how to build a better QA-TinyBERT as the future work.

## D $BERT_{SMALL}$ AS INITIALIZATION OF GENERAL TINYBERT

Initializing general TinyBERT with $BERT_{SMALL}$ is a straightforward idea. However, $BERT_{SMALL}$ would derive mismatched distributions in intermediate representations (e.g., attention matrices

Table 8: Results (dev) of baselines and TinyBERT on question answering tasks.

| System | SQuAD 1.1 | | SQuAD 2.0 | |
|---|---|---|---|---|
| | EM | F1 | EM | F1 |
| BERT$_{\text{BASE}}$ (Teacher) | 80.7 | 88.4 | 73.1 | 76.4 |
| *Small Models* | | | | |
| BERT-PKD($M$=4;$d'$=768;$d_i'$=3072) | 70.1 | 79.5 | 60.8 | 64.6 |
| DistilBERT($M$=4;$d'$=768;$d_i'$=3072) | 71.8 | 81.2 | 60.6 | 64.1 |
| TinyBERT($M$=4;$d'$=312;$d_i'$=1200) | 72.7 | 82.1 | 65.3 | 68.8 |
| *Medium-sized Models* | | | | |
| BERT-PKD ($M$=6;$d'$=768;$d_i'$=3072) | 77.1 | 85.3 | 66.3 | 69.8 |
| DistilBERT ($M$=6;$d'$=768;$d_i'$=3072) | 78.1 | 86.2 | 66.0 | 69.5 |
| TinyBERT ($M$=6;$d'$=768;$d_i'$=3072) | 79.7 | 87.5 | 69.9 | 73.4 |

Table 9: Results of different methods at pre-training state. TD and GD refers to Task-specific Distillation (without data augmentation) and General Distillation. The results are evaluated on development set.

| System | MNLI-m | MNLI-mm | MRPC | CoLA | Average |
|---|---|---|---|---|---|
| | (392k) | (392k) | (3.5k) | (8.5k) | |
| BERT$_{\text{SMALL}}$ (MLM&NSP) | 75.9 | 76.9 | 83.2 | 19.5 | 63.9 |
| BERT$_{\text{SMALL}}$ (MLM&NSP+TD) | 79.2 | 79.7 | 82.9 | 12.4 | 63.6 |
| TinyBERT (GD) | 76.6 | 77.2 | 82.0 | 8.7 | 61.1 |
| TinyBERT (GD+TD) | 80.5 | 81.0 | 82.4 | 29.8 | 68.4 |

and hidden states) with the teacher BERT$_{\text{BASE}}$ model, if without imitating the teacher's behaviors at the pre-training stage. Further task-specific distillation under the supervision of fine-tuned BERT$_{\text{BASE}}$ will disturb the learned distribution/knowledge of BERT$_{\text{SMALL}}$, finally leading to poor performances in some less-data tasks. The results in Table 9, show that the BERT$_{\text{SMALL}}$(MLM&NSP+TD) performs worse than the BERT$_{\text{SMALL}}$ in MRPC and CoLA tasks, which validates our hypothesis. For the intensive-data task (e.g. MNLI), TD has enough training data to make BERT$_{\text{SMALL}}$ acquire the task-specific knowledge very well, although the pre-trained distributions have already been disturbed.

To make TinyBERT effectively work for all tasks, we propose General Distillation (GD) for initialization, where the TinyBERT learns the knowledge from intermediate layers of teacher BERT at the pre-training stage. From the results of Table 9, we find that GD can effectively transfer the knowledge from the teacher BERT to the student TinyBERT and achieve comparable results with BERT$_{\text{SMALL}}$ (61.1 vs. 63.9), even without performing the MLM and NSP tasks. Furthermore, the task-specific distillation boosts the performances of TinyBERT by continuing on learning the task-specific knowledge of fine-tuned teacher BERT$_{\text{BASE}}$.

# E   MORE COMPLETE COMPARISONS WITH SAME STUDENT ARCHITECTURE

For the easy and direct comparisons with prior works, we here also present the results of TinyBERT ($M$=6;$d'$=768;$d_i'$=3072) with the same architectures as used in the original BERT-PKD (Sun et al., 2019) and DistilBERT[2]. Since in the original papers, the BERT-PKD is evaluated on the test set, and the DistilBERT is evaluated on the dev set. Thus, for a clear illustration, we present the results in the following Tables 10 and 11, separately.

Table 10: Comparisons between TinyBERT and BERT-PKD, and the results are evaluated on the test set of official GLUE tasks.

| System | SST-2 | MRPC | QQP | MNLI-m | MNLI-mm | QNLI | RTE |
|---|---|---|---|---|---|---|---|
| | (67k) | (3.7k) | (364k) | (393k) | (393k) | (105k) | (2.5k) |
| | Acc | F1/Acc | F1/Acc | Acc | Acc | Acc | Acc |
| *Same Student Architecture ($M$=6;$d'$=768;$d_i'$=3072)* | | | | | | | |
| BERT$_6$-PKD | 92.0 | 85.0/79.9 | 70.7/88.9 | 81.5 | 81.0 | 89.0 | 65.5 |
| TinyBERT | 93.1 | 87.3/82.6 | 71.6/89.1 | 84.6 | 83.2 | 90.4 | 66.0 |

Table 11: Comparisons between TinyBERT with DistilBERT, and the results are evaluated on the dev set of GLUE tasks. Mcc refers to Matthews correlation and Pear/Spea refer to Pearson/Spearman.

| System | CoLA (8.5k) Mcc | MNLI-m (393k) Acc | MNLI-mm (393k) Acc | MRPC (3.7k) F1/Acc | QNLI (105k) Acc | QQP (364k) F1/Acc | RTE (2.5k) Acc | SST-2 (67k) Acc | STS-B (5.7k) Pear/Spea |
|---|---|---|---|---|---|---|---|---|---|
| *Same Student Architecture ($M$=6;$d'$=768;$d'_i$=3072)* | | | | | | | | | |
| DistillBERT | 42.5 | 81.6 | 81.1 | 88.3/82.4 | 85.5 | 87.7/90.6 | 60.0 | 92.7 | 84.5/85.0 |
| TinyBERT | 54.0 | 84.5 | 84.5 | 90.6/86.3 | 91.1 | 88.0/91.1 | 70.4 | 93.0 | 90.1/89.6 |

Thus, from the direct comparisons with the reported results in the original papers, we can see the TinyBERT outperforms the baselines (DistilBERT and BERT-PKD) under the same settings of architecture and evaluation, the effectiveness of TinyBERT is confirmed. Moreover, since BERT-PKD and DistilBERT need to initialize their student models with some layers of pre-trained teacher BERT, they have the limitations that the student models have to keep the same size settings of hidden size and feedforward/filter size as their teacher BERT. TinyBERT is initialized by general distillation, so it has the advantage of being more flexible in model size selection.

