# OpenReview forum: "TinyBERT: Distilling BERT for Natural Language Understanding"
_ICLR.cc/2020/Conference — Reject_

### Official Review · AnonReviewer3 · 2019-10-18
**Official Blind Review #3**

**Rating:** 3

**Review:**

The authors propose TinyBERT, a smaller version of BERT that is trained with knowledge distillation. The authors evaluate on the GLUE benchmark.

Overall, I find the direction of this work exciting and making these large models smaller for practical use is an important research area. The authors provide various ablation experiments that provide insight into their method. The main contribution is experiments comparing various existing distillation methods to different parts of the model (embeddings, layers, prediction layer), so is not particularly novel in contributing new techniques for distillation. That being said, there is importance in contributing these results as they are very useful for others working in the area and on making smaller models. But I would expect the authors to be much more detailed in their experimental description and make it clear in the paper that the comparative baselines are fair and well tuned.

Comments:

1. Can the authors please add details for how the model has been trained, such as the datasets used, the number of update steps, the batch size, etc. as well as the finetuning parameters that were cross validated for GLUE? It is difficult to tell in the current setting if the models are comparable to the baselines. The current paper doesn't seem like it could be reproduced. It is particularly important to detail how the finetuning was done, as this is very important for the smaller datasets in GLUE.

2. Is the learning of the distilled model only done on the training dataset, or there is data augmentation beyond the training set? What is the effect without data augmentation?

3. Unfortunately, the performance drop on the GLUE benchmark as shown in Table 2 is fairly large. The authors compare to BERT Small and DistilBERT and I like the baselines, but the claim that the model achieves comparable performance to BERT Base is not true.

4. Was the BERT Small model tuned, or the same learning parameters from BERT Base were used?

5. Can the authors clarify the inference time of BERT Small? The speed improvement of TinyBERT should be the same as BERT Small based on parameter size.

6. The authors experiment with distilling the embedding layer to reduce the number of parameters, why not reduce the parameter size by reducing the vocabulary size? Existing approaches to BERT training use BPE with ~30k vocabulary size or RoBERTa with ~50k vocabulary size, but large gains could be applied here by reducing the size or using softmax reduction techniques that were popular on full vocabulary language modeling datasets like wikitext-103 or billion word.

7. Can the authors please clarify the construction of Table 2? Are those results on the test set (e.g. evaluated on the official GLUE benchmark), or on the dev set? Where are the DistilBERT numbers on the test set coming from, as it is not reported in their paper?

**Experience Assessment:**

I have published in this field for several years.

**Review Assessment: Checking Correctness Of Derivations And Theory:**

I assessed the sensibility of the derivations and theory.

**Review Assessment: Checking Correctness Of Experiments:**

I carefully checked the experiments.

**Review Assessment: Thoroughness In Paper Reading:**

I read the paper thoroughly.

---

> ### Author Response · Authors · 2019-11-13
> **Response to Reviewer #3 [1/2]**
>
> Thank you for the helpful comments!
>
> Reproducibility
>
> *** We will release the source code, all the models (including the general TinyBERT variants and task-specific TinyBERT models for each task in GLUE and SQuAD, so other researchers can easily reproduce the results in the paper), and all the training details for reproducibility, as soon as possible. ***
>
> Q1: Details for how the model has been trained.
>
> A1:
> We have presented all the training details of TinyBERT and baselines in the Appendix B (TinyBERT setup and Baselines setup), which includes the datasets used, the number of update steps, the batch size as well as the settings for fine-tuning.
>
> Our TinyBERT and baselines use the same hyper-parameters and datasets at both the pre-training and fine-tuning stages. We here list the main setting details as follows and other details can be referred in Appendix B.
>
> Table: the hyper-parameters of DistilBERT, TinyBERT and BERT_small; BERT_PKD does not include the pre-training stage, we use the BERT_base released by google as the teacher.
> -----------------------------------------------------------------------------------------------------
> 				 	        At Pre-training Stage
> -----------------------------------------------------------------------------------------------------
> Dataset		             English Wikipedia (2,500 M words)
> -----------------------------------------------------------------------------------------------------
> Training steps         		     ~350k (3epoch)
> -----------------------------------------------------------------------------------------------------
> Batch size					   256
> -----------------------------------------------------------------------------------------------------
> Learning rate			          1e-4
> -----------------------------------------------------------------------------------------------------
> Weight decay 				  1e-4
> -----------------------------------------------------------------------------------------------------
>
>
> Table: the fine-tuning hyper-parameters of DistilBERT, TinyBERT, BERT_PKD, BERT_small and BERT_base. Max_seq_length refers to the maximum sequence length.
> ---------------------------------------------------------------------------------------------------------------------------------
> 				 		Learning rate         batch size	   Epoch      Max_seq_length
> ---------------------------------------------------------------------------------------------------------------------------------
> MNLI (392k)				     3e-5				32			3                  128
> --------------------------------------------------------------------------------------------------------------------------------
> QQP (363k)				     3e-5				32			3                  128
> --------------------------------------------------------------------------------------------------------------------------------
> QNLI (108k) 			 	     3e-5				32			3                  128
> --------------------------------------------------------------------------------------------------------------------------------
> SST-2 (67k)			             3e-5				32			3                   64
> --------------------------------------------------------------------------------------------------------------------------------
> CoLA (8.5k)			             3e-5				32			3                   64
> -------------------------------------------------------------------------------------------------------------------------------
> STS-B (5.7k) 				     3e-5				32			3                   64
> -------------------------------------------------------------------------------------------------------------------------------
> MRPC (3.5k)				     3e-5				32			3                  128
> ------------------------------------------------------------------------------------------------------------------------------
> RTE (2.5k) 			             3e-5				32			3                  128
> ------------------------------------------------------------------------------------------------------------------------------
>
>
> Q2: Is the learning of the distilled model only done on the training dataset
>
> A2: As shown in the figure 2 and section 3.2, we described the methods of doing data augmentation in TinyBERT learning, the augmented dataset is merged with the original training dataset for the task-specific distillation. The effect of TinyBERT without DA (data augmentation) is presented in the Table 5 (the “No DA” row) and section 4.4.
>
>
> Q3: Claim that model achieves comparable performance to BERT_base.
>
> A3: Thanks for the suggestion, we have changed the related claims.
>
>
> Q4: Was the BERT_small model tuned, or the same learning paramters from BERT_Base were used?
>
> A4: As described in the subsection “Baselines setup” of Appendix B, BERT_small and BERT_Base use the same hyper-parameters for learning.
>
>
> Q5: Can the authors clarify the inference time of BERT Small?
>
> A5: Yes, BERT_small and TinyBERT have the same architecture, thus they have the same inference time. We have added the inference time of BERT_small in Table 3.

---

> > ### Author Response · Authors · 2019-11-13
> > **Response to Reviewer #3 [2/2]**
> >
> > Q6. Why not reduce the parameter size by the vocabulary size?
> >
> > A6: Thanks for the good suggestion. Reducing vocabulary size and reducing hidden size are two orthogonal methods to reduce the number of parameters in embedding layers. In this paper, we focused on the technique of reducing hidden size, and it would be very interesting to combine the suggested methods to further reduce the size of embedding layers.
> >
> >
> > Q7: clarify the construction of Table 2.
> >
> > A7:
> > All results in the Table 2 are evaluated on the TEST set of official GLUE benchmark. Both our TinyBERT and DistilBERT have the same number of layers 4 (M=4). The reported results of the original DistilBERT [1] are based on a 6-layer architecture and evaluated on the DEV set of GLUE tasks. And in the Table4, we compared to DistilBERT by directly using the reported results of DistilBERT paper on the DEV set, and our 6-layer TinyBERT has an average value, which is significantly better than that of DistilBERT (77.3 vs 71.9).
> >
> > As described in the Appendix B (Baselines setup), to obtain the results of 4-layer DistilBERT as reported in our paper, we firstly re-implemented the reported results of 6-layer DistilBERT with its released code to ensure our implementation procedure is correct. Then following the verified procedure, we trained a 4-layer DistilBERT as the baselines and evaluated it on the TEST set of official GLUE benchmark.
> >
> > [1] https://medium.com/huggingface/distilbert-8cf3380435b5 Accessed on 7 November 2019.

---

> > > ### Author Response · Authors · 2020-11-25
> > > **Related code has been released for more than one year**
> > >
> > > *****************
> > > As promised the related code has been released for more than one year,
> > >
> > > https://github.com/huawei-noah/Pretrained-Language-Model/tree/master/TinyBERT

---

### Official Review · AnonReviewer1 · 2019-10-21
**Official Blind Review #1**

**Rating:** 8

**Review:**

This paper proposes a new knowledge distillation method for BERT models. A number of modifications to the vanilla knowledge distillation method of Hinton et al (2015) are proposed. First, authors suggest adding L2 loss functions between alignment matrices, embedding layer values and prediction layer values. Second, authors propose run knowledge-distillation twice, once with the original pre-trained BERT model as teacher, and then again with task specific fine-tuned BERT as a new teacher. Third, authors emphasize the use of data augmentation for successful knowledge distillation. In Table 2, authors claim a significant lift across GLUE benchmarks with respect to other baseline methods with comparable model size.

While the main contribution of this paper is the proposal of empirically useful techniques than theoretical development, the empirical results reported in this paper are somewhat puzzling.

First of all, GLUE benchmark scores reported in Table 2 don't seem to be consistent with Table 1 of Sun et al (2019) for BERT-PKD ( https://arxiv.org/pdf/1908.09355.pdf ) or DistilBERT ( https://medium.com/huggingface/distilbert-8cf3380435b5 ). Indeed, BERT-PKD in Sun et al seems to significantly outperform TinyBERT on QNLI (89.0 vs 87.7) and RTE (65.5 vs 62.9), and the gap between BERT-PKD and TinyBERT on other tasks are much smaller if we take numbers reported in the original paper.

In Table 6, ablation studies with different distillation objectives are reported. Quite surprisingly, without Transformer-layer distillation (No Trm) the performance drops quite significantly. This is unexpected, because baselines such as Sun et al and DistilBERT do not use the Transformer-layer distillation but much more competitive to full TinyBERT than TinyBERT without Transformer-layer distillation. Would there be a reason why TinyBERT is so critically dependent on Transformer-layer distillation? Similarly, the removal of data augmentation (Table 5, No DA) is so detrimental to the performance of the model that it makes me to suspect whether the most of gain is from successful data augmentation. Indeed, 'No DA' row of Table 5 is very close to the performance BERT-PKD in Table 4, although the number of layers is different (4 vs 6).

In order for the research community to understand the contribution of proposed techniques more thoroughly, I suggest authors to conduct ablation studies with the simplest baseline. That is, rather than starting with the full TinyBERT model, start with a simple but competitive baseline like BERT-PKD, and only add one technique (DA, GD, Transformer-layer distillation) at a time so that readers shall understand what technique is the most important to be added to the baseline, and also whether some of the proposed techniques should always be used in combination.

---
After Author Rebuttal: authors have addressed all of my concerns quite clearly. Additional experiments which targeted a specific design choice at a time made me much more convinced that the techniques proposed in this paper are useful not only for this particular context but also more broadly applicable.

**Experience Assessment:**

I have published one or two papers in this area.

**Review Assessment: Checking Correctness Of Derivations And Theory:**

N/A

**Review Assessment: Checking Correctness Of Experiments:**

I carefully checked the experiments.

**Review Assessment: Thoroughness In Paper Reading:**

I read the paper thoroughly.

---

> ### Author Response · Authors · 2019-11-13
> **Response to Reviewer #1 [1/3]**
>
> Thank you for the helpful comments!
>
> Q1: GLUE scores reported in Table 2 don’t seem to be consistent.
>
> A1:
> In Table2, all the results of TinyBERT, BERT-PKD and DistilBERT are based on 4-layer architectures and evaluated on the TEST set of official GLUE benchmark. The results of Table 1 of Sun et al (2019) for BERT-PKD and the results of DistilBERT[1] are based on 6-layer architectures, and DistilBERT only reported their results on the DEV set of GLUE.
>
> As described in Appendix B (“Baseline setup”), to ensure the correct implementations of BERT-PKD and DistilBERT, we firstly re-reproduced the reported results of baselines with 6-layer architecture, then we trained the baselines with 4-layer architecture by following the confirmed correct implementations, and evaluated them on the TEST set of official GLUE benchmarks.
>
> For a direct comparisons with BERT-PKD and DistilBERT, we here also present the results of 6-layer TinyBERT with the same architecture as the original BERT-PKD (Sun et al., 2019) and original DistilBERT [1], and directly use the reported results of BERT-PKD and DistilBERT. As BERT-PKD and DistilBERT are evaluated on the test and dev set of GLUE, respectively. Thus, we present the results in the following two tables separately, and the results have been added to the Appendix E of our paper.
>
> Table: the comparisons between TinyBERT and BERT-PKD, and the results are evaluated on the test set of official GLUE tasks.
> -------------------------------------------------------------------------------------------------------------------------------------------------------
> 				                                SST-2         MRPC           QQP        MNLI-m    MNLI-mm    QNLI        RTE
>                                                                 (67k)          (3.7k)          (364k)        (393k)         (393k)       (105k)      (2.5k)
>                                                                  acc           f1/acc          f1/acc          acc               acc             acc          acc
> -------------------------------------------------------------------------------------------------------------------------------------------------------
> BERT_6-PKD (Sun et al., 2019)           92.0        85.0/79.9     70.7/88.9      81.5              81.0           89.0         65.5
> (M=6; d’=768;d’_i=3072)
> -------------------------------------------------------------------------------------------------------------------------------------------------------
> TinyBERT                                               93.1        87.3/82.6     71.6/89.1      84.6              83.2           90.4         66.0
> (M=6; d’=768;d’_i=3072)
> --------------------------------------------------------------------------------------------------------------------------------------------------------
>
>
> Table: the comparisons between TinyBERT with DistilBERT, and the results are evaluated on the dev set of GLUE tasks. Mcc refers to Matthews correlation and pear/spea refer to pearson/spearman.
> ---------------------------------------------------------------------------------------------------------------------------------------------------------------
> 				                 CoLA     MNLI     MNLI-mm      MRPC         QNLI        QQP         RTE      SST-2        STS-B
>                                                  mcc        acc            acc              f1/acc          acc         f1/acc        acc        acc       pear/spea
> --------------------------------------------------------------------------------------------------------------------------------------------------------------
> DistilBERT [1]                         42.5        81.6            81.1         88.3/82.4       85.5     87.7/90.6     60.0      92.7       84.5/85.0
> (M=6; d’=768;d’_i=3072)
> --------------------------------------------------------------------------------------------------------------------------------------------------------------
> TinyBERT                                54.0        84.5            84.5         90.6/86.3       91.1     88.0/91.1     70.4      93.0       90.1/89.6
> (M=6; d’=768;d’_i=3072)
> --------------------------------------------------------------------------------------------------------------------------------------------------------------
>
> [1] https://medium.com/huggingface/distilbert-8cf3380435b5 Accessed on 7 November 2019.

---

> > ### Author Response · Authors · 2019-11-13
> > **Response to Reviewer #1 [2/3]**
> >
> > Q2: Without Transformer-layer distillation, the performance drops quite significantly.
> >
> > A2:
> > The reason is about the initialization of student BERT. Based on the Table 1, we know that the ablation study of “No-Trm” (without transformer-layer distillation) as shown in Table 6, means we only do embedding-layer distillation (Embd) at the pre-training stage, and do the prediction-layer (Pred) and embedding-layer distillation (Embd) at the fine-tuning stage. At the pre-training stage, without Transformer-layer distillation, the important transformer-layers are randomly initialized and there is no supervision signal from upper layers to update their parameters during the pre-training stage.
> >
> > At the pre-training stage, (1) BERT-PKD_k are directly initialized with the first k layers of their pre-trained teacher BERT to preserve the intermediate structures of their teacher BERT; (2) DistilBERT are also firstly initialized by its teacher BERT by taking one layer out of two, then trained with the self-supervised objective over large-scale corpus to finally get a good initialization for student BERT.
> >
> > Good initialization of student BERT is very crucial for the distillation of transformer-based models in NLP tasks, and our initial motivation for general distillation (Embd+Attn+Hidn) is to make the student TinyBERT learn the intermediate structures of teacher BERT at the pre-training stage to finally get a good initialization. In the Appendix D, we also studied other initialization strategies, e.g. initializing with BERT_SMALL and obtained some interesting observations.
> >
> > Q3: ‘No DA’ row of Table 5 is very close to the performance BERT-PKD.
> > A3: The results of ‘No DA’ row in Table 5 and the results BERT-PKD in Table 4 cannot be compared directly, because they used different layer numbers, hidden sizes and feedforward/filter sizes. Thus, for a fair comparison, we evaluated the TinyBERT (No DA) with the same architecture (M=6;d’=768;d’_i=3072) as the BERT-PKD in Table 4, and all results are evaluated on DEV set and presented as follows:
> >
> > Table: the comparisons between TinyBERT (No DA) and BERT-PKD with the same architecture, and the results are evaluated on DEV set.
> > ---------------------------------------------------------------------------------------------------
> > 				                    CoLA        MNLI-m     MNLI-mm      MRPC
> >                                                      mcc             acc                acc               acc
> > ---------------------------------------------------------------------------------------------------
> > BERT-PKD
> > (M=6;d’=768; d’_i=3072)         43.1            80.9                80.9             83.1
> > ---------------------------------------------------------------------------------------------------
> > TinyBERT *No DA*
> > (M=6;d’=768;d’_i=3072)          49.1            84.0                84.4             86.0
> > ---------------------------------------------------------------------------------------------------
> > TinyBERT
> > (M=6;d’=768;d’_i=3072)          54.0            84.5                84.5             86.3
> > ---------------------------------------------------------------------------------------------------
> >
> > From the table, we can find that TinyBERT under the “No DA” setting, which performs general distillation and task-specific distillation on original training dataset, consistently outperforms the BERT-PKD with the same architecture. The results indicate that the GD (general distillation) and TD (task-specific distillation) are crucial for TinyBERT learning, and the combination of GD, TD, and DA can further increase the performances.
> >
> > Another interesting observation is that DA has bigger effect on 4-layer architecture than 6-layer architecture. We hypothesize that since the bigger 6-layer architecture has relatively larger model capacity, it can obtain better generalization capability than smaller 4-layer architecture through the pre-training over large-scale unsupervised corpus. Thus, the effect of data augmentation on the 6-layer architecture is relatively less than on the 4-layer architecture at the task-specific learning stage.

---

> > > ### Author Response · Authors · 2019-11-13
> > > **Response to Reviewer #1 [3/3]**
> > >
> > > Q4: Study the importance of the proposed techniques.
> > >
> > > A4:
> > >  Thanks for the good suggestion. Following the suggestion, we studied the importance of our proposed techniques under different experimental settings, which includes the “BERT-PKD + DA”, “BERT + GD”, “BERT + GD +TD” and the combination of “BERT-PKD + TD + GD + DA”. Note that we perform transformer distillation at both pre-training stage and fine-tuning stage, meaning that the suggested setting “BERT-PKD + transformer distillation” is equal to the setting of “BERT-PKD + GD + TD”. All the results are presented as follows:
> > >
> > > Table: the effects of different proposed techniques on the baseline BERT-PKD.
> > > ------------------------------------------------------------------------------------------------------------------------------------
> > > 				                         Architecture                   CoLA        MNLI        MNLI-mm     MRPC
> > >                                                                                                    mcc           acc                acc             acc
> > > --------------------------------------------------------------------------------------------------------------------------------------
> > > BERT-PKD                             (M=6;d’=768;d’_i=3072)          43.1           80.9              80.9            83.1
> > > --------------------------------------------------------------------------------------------------------------------------------------
> > > BERT-PKD + DA                    (M=6;d’=768;d’_i=3072)         47.7           82.8               82.9           85.5
> > > --------------------------------------------------------------------------------------------------------------------------------------
> > > BERT-PKD + GD                   (M=6;d’=768;d’_i=3072)          45.7           82.0              82.2            83.6
> > > --------------------------------------------------------------------------------------------------------------------------------------
> > > BERT-PKD                             (M=6;d’=768;d’_i=3072)         48.4           83.8               84.0            85.5
> > > Transformer-layer
> > > Distillation (GD+TD)
> > > --------------------------------------------------------------------------------------------------------------------------------------
> > > BERT-PKD+GD+TD+DA      (M=6;d’=768;d’_i=3072)         53.5            84.1               84.0            87.0
> > > --------------------------------------------------------------------------------------------------------------------------------------
> > >
> > > From the upper table, with iteratively adding the proposed techniques (DA, GD, or TD+GD) to BERT-PKD, the augmented BERT-PKD can obtain more competitive results, which are significantly better than the original BERT-PKD.
> > >
> > > “BERT-PKD + GD” achieves slightly better performances than BERT-PKD, which confirms that the proposed GD can provide a relatively better initialization to BERT-PKD. In the original BERT-PKD, it is initialized by the first 6 layers of teacher BERT-base, thus BERT-PKD has the limitations that it should have the same hidden size, feedforward / filter sizes as its teacher. In our framework, initialized by GD, so it has the advantage of being more flexible in model size selection. By continuing performing the TD, BERT-PKD (GD + TD) can further capture more task-specific knowledge and achieve better results, which demonstrates the effectiveness of two-stage learning.
> > >
> > > So we can conclude that: (a) all the proposed techniques DA, GD and transformer distillation (TD + GD) are crucial for improving the performances of baseline BERT-PKD, (b) GD can provide a good initialization to BERT-PKD, (c) we can get the best performances with combining all the proposed techniques GD+TD+DA.

---

> > > > ### Comment · AnonReviewer1 · 2019-11-14
> > > > **Good clarifications**
> > > >
> > > > Thanks a lot for clarifications. I realized that I missed some of the important details in experiments such as the difference of hidden unit sizes/# of layers or how student layers were initialized. These experiments have quite convincingly resolved concerns I raised, and I will reflect my score accordingly.

---

### Official Review · AnonReviewer2 · 2019-10-29
**Official Blind Review #2**

**Rating:** 6

**Review:**

What is the task?
Knowledge distillation of BERT

What has been done before?
Unlike prior works such as Distilled BiLSTMSOFT (Tang et al., 2019), BERT-PKD (Sun et al., 2019) and DistilBERT, this work

i) Do knowledge distillation at pre training stage also in addition to fine tuning stage.
ii) Student learns from all - embedding layers, attention matrices, hidden states, and final prediction layers.

In BERT-PKD, student learns from the [CLS]  hidden states of the teacher.

What are the main contributions of the paper?
Novel Transformer distillation method that is specially designed for knowledge distillation of the Transformer-based models.
Novel two-stage learning framework which performs Transformer distillation at both the pre-training and task-specific learning stages
Resulting TinyBERT being 7.5x smaller and 9.4x faster on inference and significantly outperforms other state-of-the-art baselines on BERT distillation.

What are the key techniques used to tackle this task?
Novel Transformer distillation method that is specially designed for knowledge distillation of the Transformer-based models.
Novel two-stage learning framework which performs Transformer distillation at both the pre-training and task-specific learning stages

What are the main results? Are they significant?
Resulting TinyBERT being 7.5x smaller and 9.4x faster on inference and significantly outperforms other state-of-the-art baselines on BERT distillation with only ∼28% parameters and ∼31% inference time of them.

Results show that three key procedures: TD (Task-specific Distillation), GD (General Distillation) and DA (Data Augmentation) are crucial for the proposed KD method.

Proposed distillation objectives - Transformer-layer distillation (attention matrices and hidden states), embedding-layer distillation and prediction layer distillation  are crucial for the proposed KD method.

Weaknesses
experimental results were not easily comparable to prior work so it is hard to say if claims are well-supported experimental results

Questions
Did authors try other values of lambda


**Experience Assessment:**

I have published one or two papers in this area.

**Review Assessment: Checking Correctness Of Derivations And Theory:**

N/A

**Review Assessment: Checking Correctness Of Experiments:**

I carefully checked the experiments.

**Review Assessment: Thoroughness In Paper Reading:**

I read the paper at least twice and used my best judgement in assessing the paper.

---

> ### Author Response · Authors · 2019-11-13
> **Response to Reviewer #2 [1/2]**
>
> Thank you for the helpful comments!
>
> Q1: Experimental results are not easily comparable to prior work.
>
> A1:
> *** Comparison results as shown in Table 2, Table 3 and Table4 ***
> The comparison results as shown in the Table 2 are all evaluated on the test set of the official GLUE tasks. As shown in the Table 3, our TinyBERT, baselines BERT-PKD and DistilBERT, all have the same number of layers (M=4), and our TinyBERT has a relatively challenging setting with smaller hidden size (d’=312 vs d’=768) and feedforward/filter size (d’_i=1200 vs d’_i=3072). If we increase the hidden size and feedforward/filter size of TinyBERT, it can obtain better performances, which is validated in our experiments in the Table 4 (wider TinyBERT variants achieve better results).
>
> In the Table 4, we also directly compared the performances of TinyBERT, BERT-PKD and DistilBERT with the same architecture settings (M=6; d’=768; d’i=3072), and TinyBERT has significantly better performances.
>
> ***More complete comparisons with the same student architecture ***
> For complete and direct comparisons with prior works, we here also present the results of TinyBERT (M=6; d’=768; d’_i=3072) with the same architectures as used in the original BERT-PKD (Sun et al., 2019) and DistilBERT [1] papers.
>
> Since in the original papers, the BERT-PKD is evaluated on the TEST set, and the DistilBERT is evaluated on the DEV set. Thus, for a clear illustration, we present the results in the following two tables, separately, and the results have been added to the Appendix E of our paper.
>
> Table: the comparisons between TinyBERT and BERT-PKD, and the results are evaluated on the test set of official GLUE tasks.
> -------------------------------------------------------------------------------------------------------------------------------------------------------
> 				                                SST-2         MRPC           QQP        MNLI-m    MNLI-mm    QNLI        RTE
>                                                                 (67k)          (3.7k)          (364k)        (393k)         (393k)       (105k)      (2.5k)
>                                                                  acc           f1/acc          f1/acc          acc               acc             acc          acc
> -------------------------------------------------------------------------------------------------------------------------------------------------------
> BERT_6-PKD (Sun et al., 2019)           92.0        85.0/79.9     70.7/88.9      81.5              81.0           89.0         65.5
> (M=6; d’=768;d’_i=3072)
> -------------------------------------------------------------------------------------------------------------------------------------------------------
> TinyBERT                                               93.1        87.3/82.6     71.6/89.1      84.6             83.2            90.4         66.0
> (M=6; d’=768;d’_i=3072)
> --------------------------------------------------------------------------------------------------------------------------------------------------------
>
>
> Table: the comparisons between TinyBERT with DistilBERT, and the results are evaluated on the dev set of GLUE tasks. Mcc refers to Matthews correlation and pear/spea refer to pearson/spearman.
> ---------------------------------------------------------------------------------------------------------------------------------------------------------------
> 				                 CoLA     MNLI     MNLI-mm      MRPC         QNLI        QQP         RTE      SST-2        STS-B
>                                                  mcc        acc            acc              f1/acc          acc         f1/acc        acc        acc       pear/spea
> --------------------------------------------------------------------------------------------------------------------------------------------------------------
> DistilBERT [1]                        42.5        81.6            81.1         88.3/82.4       85.5     87.7/90.6     60.0      92.7       84.5/85.0
> (M=6; d’=768;d’_i=3072)
> --------------------------------------------------------------------------------------------------------------------------------------------------------------
> TinyBERT                               54.0        84.5            84.5         90.6/86.3       91.1     88.0/91.1     70.4      93.0       90.1/89.6
> (M=6; d’=768;d’_i=3072)
> --------------------------------------------------------------------------------------------------------------------------------------------------------------

---

> > ### Author Response · Authors · 2019-11-13
> > **Response to Reviewer #2 [2/2]**
> >
> > Thus, from the direct comparisons with the reported results in the original papers, we can see the TinyBERT outperforms the baselines (DistilBERT and BERT-PKD) under the same settings of architecture and evaluation, the effectiveness of TinyBERT is confirmed.
> >
> > Moreover, since BERT-PKD and DistilBERT need to initialize their student models with some layers of pre-trained teacher BERT, they have the limitations that the student models have to keep the same size settings of hidden size and feedforward/filter size as their teacher BERT. TinyBERT is initialized by general distillation, so it has the advantage of being more flexible in model size selection.
> >
> > [1] https://medium.com/huggingface/distilbert-8cf3380435b5 Accessed on 7 November 2019.
> >
> >
> >
> > Q2: Did authors try other values of lambda.
> >
> > A2: Yes, we have tried a different strategy at the early stage, which is formatted as $\lambda_{m}=0.2*(1 + m), 0<=m<=M$, assigning larger weights to higher layers. The results are presented in the following table and show that the performances obtained by the proposed strategy on datasets CoLA, MNLI-m/mm and MRPC are worse than the ones obtained by the uniform strategy ($\lambda_{m}=1$). We also find the knowledge distillation in CV often uses the uniform strategy [Romero et al., 2014], thus we move to the uniform strategy.
> >
> > Table: the comparisons between different values of lambda and the results are evaluated on dev set.
> > ----------------------------------------------------------------------------------------------------
> > 				                        CoLA       MNLI-m      MNLI-mm       MRPC
> >                                                          mcc          acc                 acc                 acc
> > ----------------------------------------------------------------------------------------------------
> > TinyBERT                                        49.7          82.8               82.9                85.8
> > (M=4; d’=312;d’_i=1200)
> > $\lambda_{m}=1$
> > ---------------------------------------------------------------------------------------------------
> > TinyBERT                                       43.7          80.7               81.4                 83.8
> > (M=4; d’=312;d’_i=1200)
> > $\lambda_{m}=0.2*(1 + m)$
> > ---------------------------------------------------------------------------------------------------

---

### Public Comment · ~Xianghao_Tang1 · 2019-09-29
**The ablation study on Transformer-layer distillation**

The general distillation doesn't use prediction-layer distillation loss. Why? The pre-training objectives seem also suitable for it.
In ablation study, if Transformer-layer distillation is removed in the general distillation, the student only learns from the teacher's embedding layer and can learn nearly nothing, which definitely hurts the pre-training of student a lot and is not fair.

---

> ### Author Response · Authors · 2019-09-30
> **Why not conducting prediction-layer distillation at pre-training stage**
>
> Thanks for your comments!
> In the proposed two-stage learning framework, four different distillations (including the prediction-layer distillation) are considered at both the pre-training and fine-tuning stage, more details can be found in Table-1. Our initial motivation for general distillation is to make the student TinyBERT learn the intermediate structures of teacher BERT at the pre-training stage. Moreover, from our preliminary experiments, we also found that conducting “prediction-layer distillation” at the pre-training stage would not bring extra improvements on downstream tasks, when the Transformer-layer distillation (Attn and Hidn distillation) and Embedding-layer distillation have already been performed. Therefore, we did not include the prediction-layer distillation at the pre-training stage and this setting is followed in the ablation study.

---

### Author Response · Authors · 2019-11-13
**Summary of Submission Changes (1/10/2020)**

#Update(1/10/2020)
===============================================================================================
We have released our code and models at the link  https://github.com/huawei-noah/Pretrained-Language-Model/tree/master/TinyBERT and welcome to use them. Our 6-layer TinyBERT (BERT-base as teacher) can obtain almost same performances with BERT-base: GLUE score 79.4 vs 79.5.


#Update(12/2/2019)
===============================================================================================

We would like to thank the reviewers for the helpful comments!

We updated a new version including the changes as follows:

1.We added more complete comparisons with same student architecture in the Appendix E of new version, for the easy and direct comparisons with prior works.

2.We rephrased some claims more precisely and added the inference time of BERT$_{SMALL}$ based on the reviewers’ suggestions.

---

### Decision · Program_Chairs · 2019-12-19

**Decision:**

Reject

**Comment:**

This paper proposes a new distillation-based method for using large pretrained models like BERT to produce much *smaller* fine-tuned target-task models.

This paper is low-borderline: It has merit and meets our basic standards, but owing to capacity limitations we had to give preference to papers we see as having a higher potential impact. Reviewers had some concerns about experimental design, but those seem to have been fully resolved after discussion. Reviewers were not convinced, even after some discussion, that the method and results were sufficiently novel and effective to have a substantial impact on the state of practice in this area.